# Comparison between Cylindrical, Trigonal, and General Symmetry Models for the Analysis of Polarization-Dependent Second Harmonic Generation Measurements Acquired from Collagen-Rich Equine Pericardium Samples

**Meropi Mari** [1] [iD]**, Vassilis Tsafas** [1]**, Despina Staraki** [1,2]**, Costas Fotakis** [1,2] **and George Filippidis** [1,*] [iD]

[1] Institute of Electronic Structure and Laser (IESL), Foundation for Research and Technology–Hellas (FORTH), 71110 Heraklion, Greece; mmari@iesl.forth.gr (M.M.); tsafas@iesl.forth.gr (V.T.); ph4319@edu.physics.uoc.gr (D.S.); fotakis@iesl.forth.gr (C.F.)

[2] Department of Physics, University of Crete, 71003 Heraklion, Greece

[*] Correspondence: filip@iesl.forth.gr; Tel.: +30-281-039-1320

**Abstract:** Polarization-dependent second harmonic generation (PSHG) microscopy is used as an innovative, high-resolution, non-destructive, and label-free diagnostic imaging tool to elucidate biological issues with high significance. In the present study, information on the structure and directionality of collagen fibers in equine pericardium tissue was collected using PSHG imaging measurements. In an effort to acquire precise results, three different mathematical models (cylindrical, trigonal, and general) were applied to the analysis of the recorded PSHG datasets. A factor called the "ratio parameter" was calculated to provide quantitative information. The implementation of the trigonal symmetry model to the recorded data led to the extraction of improved results compared with the application of the widely used cylindrical symmetry model. The best outcome was achieved through the application of the general model that does not include any kind of symmetry for the data processing. Our findings suggest that the trigonal symmetry model is preferable for the analysis of the PSHG datasets acquired from the collagenous tissues compared with the cylindrical model approach although an increased computational time is required.

**Keywords:** second harmonic generation imaging; polarization-dependent measurements; collagen-rich samples; trigonal model; cylindrical model; data processing

## 1. Introduction

Collagen is the main structural protein in the extracellular matrix and in connective tissue. It is the most abundant protein in mammals. In this study, equine pericardium tissue, which consists of a well-organized type I collagen network, was examined. Type I collagen is present in various tissues, including muscles, cornea, bones, and teeth [1]. It has one or multiple fibrils, each one consisting of three polypeptide α-chains held together with hydrogen bonds to form a collagen helix [2]. Collagen is considered a biomarker for many important diseases, including cancer [3,4]. The development of novel, rapid, label-free, and non-invasive methods for sub cellular collagen detection and monitoring is imperative in order to achieve improved, faster, and more accurate delineation of collagen–rich tissues. Collagen fibrils consist of a triple polypeptide helix (a non-centrosymmetric structure) that comprises an ideal emitter of high second harmonic generation (SHG) signals [5].

SHG is a non-linear and coherent scattering phenomenon. This non-destructive technology enables high-resolution imaging capabilities from deep sample layers and provides intrinsic three-dimensional optical sectioning as well as prolonged periods of irradiation of the unstained specimens [6]. Polarization-dependent SHG (PSHG) imaging is an optical microscopy technique capable of quantifying molecular structural changes

occurring even below the diffraction limit [7]. PSHG imaging has been used to characterize the collagen distribution at the submicron level in tissues [8].

PSHG measurements have been previously employed to investigate the structural alterations in the collagen in the presence of malignancy [9–11]. Furthermore, this technique has been applied as a diagnostic tool for in vivo collagen structural imaging studies [12]. The analysis performed on the PSHG data is based on the cylindrical symmetry model in the majority of the approaches [11–14]. However, other works demonstrated that the molecular symmetry of collagen is better represented by the trigonal symmetry model [9,15]. In the current study, three different mathematical models were applied to analyze PSHG datasets derived from equine pericardium tissue samples. These collagen tissue grafts are commonly used as standard samples and they are able to provide high SHG signals. Applications of equine pericardium tissue include the manufacturing of heart valve prostheses and wound healing [16,17]. Moreover, it has been used to monitor changes in chemical and mechanical properties of tissue, upon collagen degradation, by employing various label-free optical imaging techniques [18].

Our objective is to compare the three different PSHG data analysis approaches and identify the model that renders the best results. Thus, a Discrete Fourier Transform (DFT) algorithm was employed for fast SHG signal analysis based on the classical cylindrical symmetry model. In addition, a trigonal symmetry model as well as a general mathematical model that is not based on symmetries were used for the PSHG data analysis. The results of our study indicate that the application of the trigonal symmetry model on the analysis of the extracted PSHG data provides significantly improved outcomes compared with the cylindrical model's application. The best results were obtained via the implementation of the general model.

## 2. Materials and Methods

### 2.1. Biological Sample

Unstained tissues from equine pericardium were studied by employing non-linear imaging measurements. Decellularized equine pericardium tissue (approximate dimensions: 10 cm × 2 cm) was provided by Auto Tissue. This collagenous tissue sample produces high SHG signals. The tissue was divided into 4 smaller pieces with approximately 1 cm × 1 cm dimensions each. For the performance of PSHG measurements, these smaller sections were placed on thin (0.07 mm) round glass coverslips (Ø 3.5 cm). The sample thickness was 0.5 mm. For the collection of the 2D PSHG measurements, a layer at a depth of ~100 μm from the sample surface was investigated.

Prior to their irradiation, the samples were washed in a phosphate-buffered saline (PBS) solution to remove any residues of the antibiotic solution in which they were stored. A few droplets of PBS solution were also added on the sample before the experiment in order to maintain tissue hydration during scanning. Multiple regions of each piece of the sample were scanned.

### 2.2. Experimental Apparatus

The experimental setup was similar to the one described in our previous studies [6,11]. Briefly, an Yb-based femtosecond (fs) laser oscillator, emitting at a central wavelength of 1028 nm (Amplitude, Bordeaux-France, 200 fs, 50 MHz), was employed as an excitation source. The laser beam was guided to a modified upright microscope (Nikon Europe, Amsterdam-Netherlands). A set of galvanometric mirrors (Cambridge Technology, Bedford-MA-USA) were utilized for the xy raster scanning of the sample. A zero-order half-wave retardation plate (WPH05ME; Thorlabs, Bergkirchen-Germany) was placed into a motorized rotation stage in order to control the orientation of the incident linear polarization. This is a prerequisite for the creation of the PSHG datasets that were analyzed using the three different models. The extinction ratio, calculated by cross polarization measurements at the sample plane, was higher than 25:1 for all linear polarization orientations. The energy per pulse at the sample plane was 0.8 nJ. The focal plane was adjusted by using a motorized xyz

translation stage (Standa Ltd., Vilnius-Lithuania). A moderate numerical aperture objective lens (Carl Zeiss, Jena-Germany, C-Achroplan 32×, NA 0.85) was used to tightly focus the beam onto the sample. The bright field observation of the specimen was performed through a CCD camera (PixeLINK). SHG signals were collected by a second objective lens (Carl Zeiss, PlanNeofluar, 40×, NA 0.8, air immersion) and detected by a photomultiplier tube (PMT Hamamatsu, H9305-04, Tokyo-Japan) in the forward direction. A bandpass interference filter (Semrock 514nm, IDEX Health & Science, LLC, Rochester-NY-USA) and a short pass filter (Semrock 720nm) were placed in front of the PMT slot to cut off the transmitted laser light and solely detect the SHG signals arising from the samples.

Our setup scans a 2D (500 × 500 pixels) SHG image in one second. Each pixel of the acquired image is a square with a 0.18 μm side dimension. To improve the signal-to-noise ratio (SNR), 10 scans were averaged for each final image. For each investigated area of the sample, 18 averaged 2D SHG images were recorded, rotating the linear polarization of the incident beam by 10 degrees each time (0–170°). These images constitute a complete PSHG dataset computed for all the three different models. By using this configuration, sufficient data for the analysis were extracted while the irradiation time of the biological sample was constrained. We have to mention that for these 18 different linear polarizations of the incident beam at the sample plane, the entire emitted SHG signal was collected.

### 2.3. PSHG Models and Data Analysis

The modulation of the SHG signal produced by the collagenous tissue samples, with respect to the rotation of the incident linear polarization of the excitation beam, when assuming that collagen fibrils are arranged in a cylindrically symmetric distribution along the fiber's axis and parallel to it, is described by the following equation [19,20]:

$$I_{SHG} = E \cdot \left\{ (\sin[2(a - f)])^2 + [(\sin(a - f))^2 + B \cdot (\cos(a - f))^2]^2 \right\} \quad (1)$$

where $E$ is an overall multiplication factor, $f$ denotes the angle between the initial polarization of the laser beam and the projection of the collagen fiber onto the polarization plane, and $\alpha$ is the rotation angle of the laser's linear polarization induced by the half-wave plate. The factor $B$, which is an indication of the collagen structure's organization, is called the anisotropy parameter.

Equation (1) can be rewritten as [21]:

$$I_{SHG} = c_0 + c_2 \cdot \cos(2 \cdot (a - f)) + c_4 \cdot \cos(4 \cdot (a - f)) \quad (2)$$

where

$$B = \sqrt{\frac{c_0 + c_2 + c_4}{c_0 - c_2 + c_4}} \quad (3)$$

The form of Equation (2) enables the calculation of the $c_0$, $c_2$, and $c_4$ coefficients (therefore of $B$) and angle $f$ through a Discrete Fourier Transform (DFT) of $I_{SHG}$ values for several different polarization angles $\alpha$ [21]. The computational time of the DFT for 500 × 500 pixels images is four orders of magnitude faster as compared with the computation of a non-linear fitting algorithm for Equation (1) that calculates the anisotropy parameter $B$ and angle $f$ values. Thus, the time needed for data processing of an SHG image with the DFT algorithm is only 1 s. The most general form of $I_{SHG}$ dependence with respect to the angle $a$ is [22,23]

$$I_{SHG} = b_0 + b_2 \cdot \cos(2 \cdot a) + b_4 \cdot \cos(4 \cdot a) + d_2 \cdot \sin(2 \cdot a) + d_4 \cdot \sin(4 \cdot a) \quad (4)$$

which can be written as

$$I_{SHG} = b_0 + b_2' \cdot \cos(2 \cdot (a - f_2)) + b_4' \cdot \cos(4 \cdot (a - f_4)) \quad (5)$$

The main differences of Equation (5) compared with Equation (2) are that no symmetry axis has been assumed and angles $f_2$ and $f_4$ (spectral phases) may differ.

If SHG emitters are arranged with the polar trigonal symmetry $3m$, then the generated SHG signal as a function of the previously defined angles $a$ and $f$, takes the following form [9].

$$I_{SHG} = E\{(\chi_{22} \cdot (\sin(a - f))^2 + \chi_{15} \cdot \sin[2(a - f)])^2 + [\chi_{31} \cdot (\sin(a - f))^2 + \chi_{33} \cdot (\cos(a - f))^2]^2\} \quad (6)$$

In the case where $\chi_{22}$ equals zero and the sample does not absorb the incident radiation (thus the Kleinman symmetry is valid and $\chi_{15}$ is equal to $\chi_{31}$), Equation (6) takes the form of Equation (1). That means that cylindrical symmetry is a specific case of trigonal symmetry and the ratio of the absolute value of $\chi_{22}$ divided by $\chi_{31}$ comprises a measure of the balance between these two symmetries [15].

The analysis of the collected SHG images with the three aforementioned mathematical models is described below. Initially, a 3D matrix (PSHG matrix) with dimensions of $500 \times 500 \times 18$ was produced from eighteen 2D SHG images of the same area, rotating each time the linear polarization of the incident laser beam by ten degrees ($\alpha = 0$–$170°$). The first two dimensions of the PSHG matrix refer to the location of the pixels from an SHG image, while the third dimension is the value of each pixel for all laser polarizations. For the cylindrical symmetry model for each pixel area ($500 \times 500$), the values of angle $f$ and coefficients $c$ from Equation (2) and, subsequently, the value of $B$ through Equation (3) were calculated via the application of DFT analysis along the third dimension of the PSHG matrix.

In particular, a specially constructed algorithm, which was designed and programmed in Python, computed the coefficients $c_0$ and $c_2$ and the angle $f$ of the left cosine term of Equation (2) by applying the DFT along the third dimension of the PSHG matrix. Afterwards, the algorithm computed the last coefficient $c_4$ by assuming as a constant the already-known value for angle $f$ of the right cosine term.

The average values for the anisotropy parameter $B$ ($<B>$) were calculated after the elimination of erroneous pixels, such as noise pixels, that were not in line with Equation (2). In this study, a threshold value was used as a filtering criterion for the determination of the coefficient $R^2$, as referred to in the literature [24]. This coefficient compares the experimental data and the predicted ones from Equation (2) after the calculation of its parameters with DFT analysis. Pixels below the threshold value were excluded from further processing. Specifically, the threshold value of 0.90 was set, and it was the maximum value that ensured that the results fit into the model, while there is a sufficient number of remaining pixels for the statistical analysis.

Moreover, the results of the aforementioned DFT analysis contain coefficients $b_4$ and $d_4$ of Equation (4) of the general model, providing thus the values for coefficient $b'_4$ and angle $f_4$ of Equation (5). There is no need to estimate the values of the coefficients $b_0$ and $b'_2$ or the angle $f_2$ of Equation (5) since the way they are calculated is identical to that of $c_0$ and $c_2$ and angle $f$ of Equation (2). However, it is worth mentioning that the recalculation of $R^2$ values is necessary.

The equation of trigonal symmetry (Equation (6)) can be written in the following form [6]:

$$I_{SHG} = b_0 + b'_2 \cdot \cos(2 \cdot (a - [f - \Delta f_2])) + b'_4 \cdot \cos(4 \cdot (a - [f - \Delta f_4])) \quad (7)$$

It is feasible, by using this formula, to calculate the coefficients $b_0$, $b'_2$, $b'_4$, $f_2$, and $f_4$ via DFT analysis. However, this type of analysis cannot confidently provide information about Equation (6) parameters ($E$, $f$, $\chi_{22}$, $\chi_{15}$, $\chi_{31}$, and $\chi_{33}$) [6]. Thus, the trigonal symmetry approach was chosen to complete the analysis by fitting the SHG modulation on Equation (6) for all the pixel areas of each sample.

Furthermore, in the frame of this study an extra factor called the "ratio parameter" was calculated and contributed additional quantitative information through PSHG analysis of the collected data. The value of this parameter was defined as the number of collagen pixels with an $R^2 > 0.9$ divided by the total number of collagen pixels. As countable pixels

arising from collagen, we considered only those that presented the minimum value along the third dimension of the PSHG matrix over the system noise. This factor provides a quantitative index of the ability of the extracted results to sufficiently describe the PSHG data of the collagenous tissue samples for the three different models (cylindrical, trigonal, and general).

## 3. Results and Discussion

Microscopic imaging of biological structures and processes is the central technology platform in life sciences and has contributed to more groundbreaking technologies than any other biomedical technology [25–27]. There is a need to understand the origin of the non-linear response of the collagen triple helix in order to perform quantitative SHG imaging of collagenous tissues. A study demonstrated that the SHG signal from type 1 collagen arises from the tight alignment of a large number of small and weakly efficient harmonophores that presumably correspond to the peptide bonds [28].

In an effort to delineate the sub-micron morphology and directionality of collagen without the need for staining, equine pericardium tissue was studied. We attempted to visualize and obtain qualitative information on collagen structures by collecting SHG signals. Figure 1 presents a 3D image from the pericardium sample. High SHG signals were detected from the collagen, indicating its organization and distribution within the tissue at a high resolution. Furthermore, working under constant irradiation conditions (mean energy per pulse of the incident beam at the sample plane, dimensions of the scanning region, number of pixels, amplification of the PMT unit), the brightness of the pixels from each image of the sample can provide additional information about the collagen concentration.

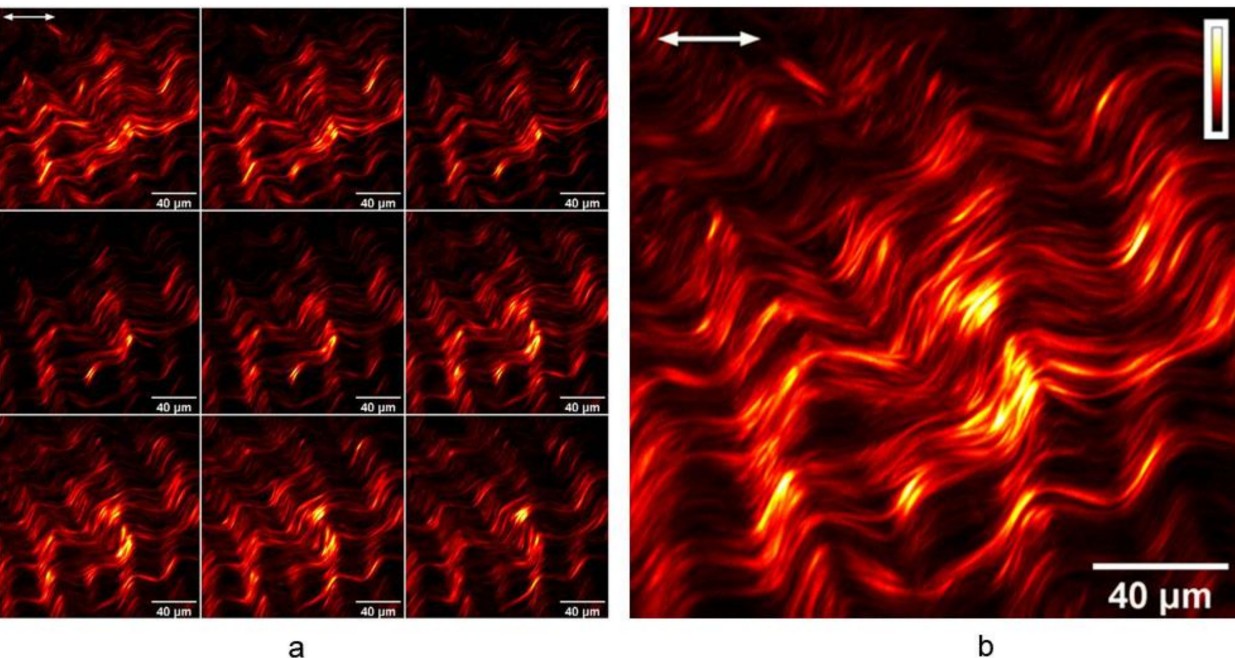

**Figure 1.** (**a**) A 3D SHG image from a collagen-rich sample. Sequential optical sections of 2 μm apart were recorded. (**b**) A z-projection to the maximum intensity of all the 9 slices of (**a**). The arrow indicates the incident horizontal laser polarization. The calibration bar denotes the maximum (white) and minimum (black) SHG intensity signals.

PSHG measurements were performed on these collagen-rich specimens and three different mathematical models for the data analysis were assessed in an attempt to determine the best approach in order to obtain precise sub cellular information. Figure 2 depicts a montage consisting of nine out of the eighteen PSHG dataset images from an equine pericardium patch. Each image of the montage represents the same area and the brightness

is propotional to the recorded SHG signals. It can be observed that the lower brightness of the collagen fibers was detected when the incoming polarization was perpendicular to the orientation of the fibers (Figure 2g). These findings are in perfect agreement with other studies that present PSHG signal modulation from different collagen-rich tissues [29].

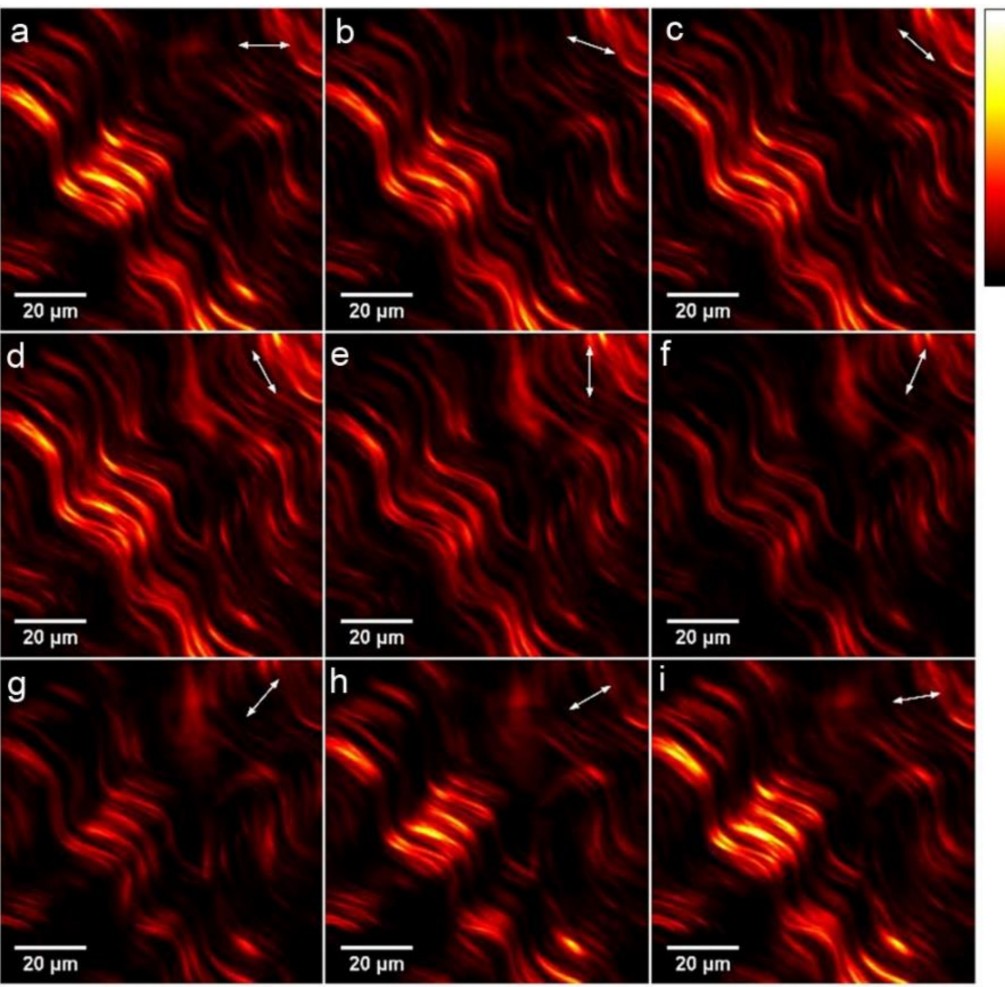

**Figure 2.** Polarization-dependent 2D SHG measurements recorded from an equine pericardium tissue. Nine out of the eighteen PSHG dataset images are shown. Every image (**a**–**i**) corresponds to the different linear polarization of the incident laser beam to the sample plane, indicated by the white arrow. The calibration bar denotes the maximum (white) and minimum (black) SHG intensity signals.

In all cases, the scanning region was limited to 90 μm × 90 μm. The number of pixels of each image was 250,000. Twenty PSHG images from different tissue regions were scanned and analyzed. For the cylindrical symmetry case, the mean value of B (<B>) as well as the standard deviation (SD) of it was estimated to be 1.57 ± 0.07, derived from pixels with an $R^2 > 0.9$. To the best of our knowledge, this is the first time that a B value has been calculated for an equine pericardium sample.

In addition, the value of <B> corresponds to an effective angle of $\theta_e$ [11] around 48.50° ± 0.57°. Previous studies on X-ray diffraction experiments evaluate the collagen helix angle at 45° [29].

To further analyze our data, three different mathematical models (cylindrical, trigonal, and general) were applied to the recorded PSHG measurements. As stated previously, for the cylindrical and general models, the DFT analysis was applied. The fitting analysis was also tested for the cylindrical symmetry, but revealed similar results to the DFT method. The results for the trigonal symmetry model were acquired by fitting the SHG modulation signal on Equation (6). By employing the trigonal symmetry analysis, the mean values of B

(<B>) as well as the SD were $1.98 \pm 0.13$. These values were calculated including solely the pixels with an $R^2 > 0.9$. By comparing the application of the trigonal and cylindrical models, it was noted that the number of pixels with an $R^2 > 0.9$ is increased for the trigonal symmetry approach. The best outcome arises when the general model is implemented. This increase in the number of pixels can be expressed through the calculation of the ratio parameter values, defined in the previous section, for the three models. This factor, via its application to the analysis (fitting or DFT), provides quantitative information on the application of each model to the obtained PSHG datasets. Specifically, higher ratio values indicate the increased ability of the model in obtaining results that sufficiently describe the PSHG data extracted from the biological collagenous sample.

Figure 3 presents the mean ratio values for the cylindrical, trigonal, and general models for the PSHG measurements recorded from the equine pericardium tissue. Figure 3 demonstrates that the lowest value of the ratio parameter was calculated for the cylindrical symmetry model. A higher ratio parameter value was computed for the trigonal symmetry model. Thus, the trigonal model is preferable for the description of the collagenous tissue in comparison with the classical cylindrical symmetry one. Moreover, Figure 3 shows that the collagen distribution is better-defined, with the highest ratio value, by employing analysis based on the general approach that does not take into account any kind of symmetry.

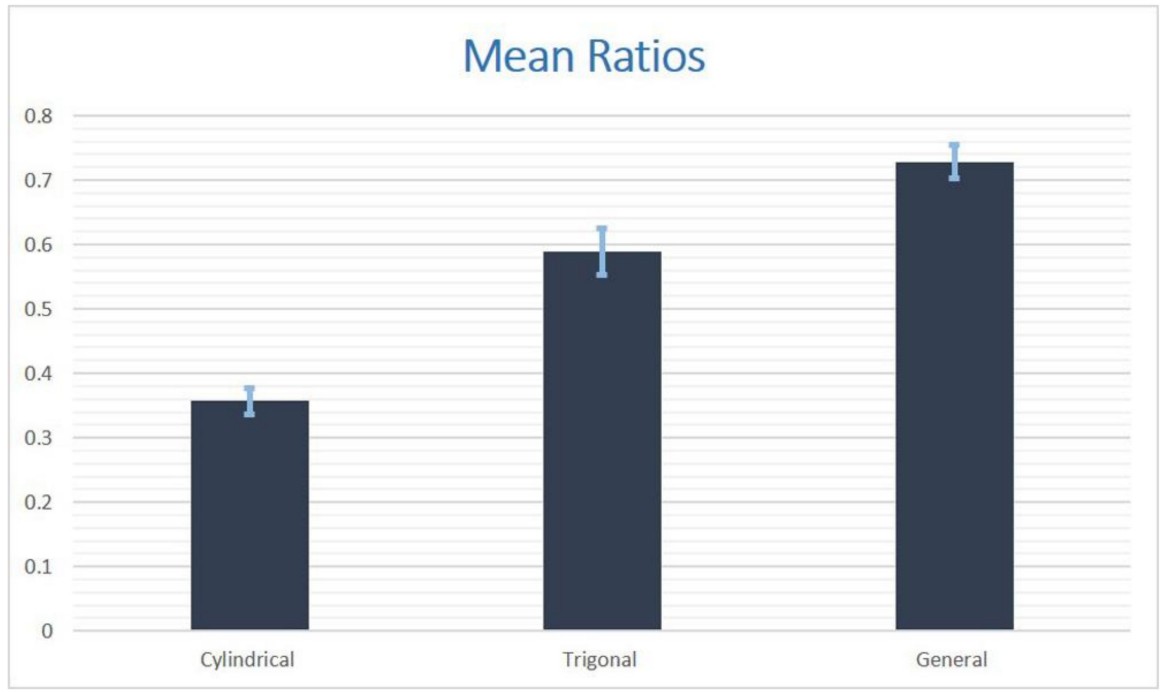

**Figure 3.** Calculated mean ratio parameter values for the cylindrical, trigonal, and general mathematical models. The error bars are the standard error of the mean (SEM) for each model. *n* = 20 measurements from four different samples.

The technology of PSHG to monitor the structural modifications of collagen at the sub cellular level is expected to become dominant not only for basic research but for application-driven studies as well in the following years. The potential application of this optical technique in preclinical and/or clinical studies presupposes the optimal analysis of the extracted data. In our study, three different models were applied to PSHG images recorded from equine pericardium tissue. The obtained results suggest that the general model renders the best outcome. Nevertheless, this model characterizes specimens with an unknown structure. From Figure 3, it can be seen that the comparison between the cylindrical model and the trigonal model leads to a significant difference in ratio parameter values. Thus, our work reinforces and proves recent findings [15] that the trigonal approach

has the potential to provide improved PSHG results from collagen-rich tissues and it should be preferred for data processing.

### 4. Conclusions

PSHG optical imaging techniques have been used as label free, non-invasive tools to reveal biological structures and activities at the sub cellular level [30]. In the present study, three different mathematical models were employed in an effort to improve the extracted PSHG results obtained from a collagen-rich tissue sample. The application of the trigonal symmetry model to the PSHG datasets, based on quantitative criteria (ratio parameter values), provides significantly improved results in comparison with the widely used cylindrical model approach. The findings presented herein indicate that the trigonal symmetry model describes more precisely the collagen distribution and organization in tissues and it is more suitable in comparison with the cylindrical one for the analysis of PSHG datasets collected from the collagenous samples. Recently, it was shown that the mechanical stretching of collagen further aligns the fibrils with the fiber axis during cancer progression and increasing anisotropy parameter B values [11]. That work was based on the application of the cylindrical symmetry model to the PSHG data. We anticipate that the application of the trigonal model will be more precise for collagen-rich tissue discrimination via the calculation of quantitative factors such as anisotropy parameter values.

The best fitting outcome was achieved via the implementation of the general model. This approach describes samples with an unknown structure. Thus, future studies should investigate the employment of even more complicated models that involve more symmetries than the trigonal one to achieve the optimal outcome from the PSHG data analysis. In addition, the differentiation of healthy from pathological tissues based on the non-linear collagen response comprises another main future research target.

**Author Contributions:** In the present study, the authors contributed to the following: conceptualization, C.F. and G.F.; methodology, M.M. and V.T.; software, V.T.; validation, M.M., V.T. and G.F.; formal analysis, M.M., V.T. and D.S.; data curation, M.M. and V.T.; writing—original draft preparation, G.F.; writing—review and editing, M.M. and V.T.; supervision, G.F. All authors have read and agreed to the published version of the manuscript.

**Funding:** The present work was supported by "LASERLAB EUROPE V" (871124). In addition, M. Mari acknowledges the Hellenic Foundation for Research and Innovation (HFRI) and the General Secretariat for Research and Innovation (GSRI) for the financial support under grant agreement No 1357.

**Institutional Review Board Statement:** Not applicable.

**Informed Consent Statement:** Not applicable.

**Data Availability Statement:** The data that support the findings of this study are available from the corresponding authors upon reasonable request.

**Conflicts of Interest:** The authors declare no conflict of interest.

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
