# Peer review of "Comparison between Cylindrical, Trigonal, and General Symmetry Models for the Analysis of Polarization-Dependent Second Harmonic Generation Measurements Acquired from Collagen-Rich Equine Pericardium Samples"

_photonics, doi:10.3390/photonics9040254_

Round 1

Reviewer 1 Report

Comparison between cylindrical and trigonal symmetry models for the analysis of polarization-dependent second-harmonic generation measurements acquired from collagen-rich biological samples

Summary: Polarization-resolved SHG data of collagen fibres in equine pericardium tissues are mathematically treated from 3 different models (cylindrical, trigonal and general) to derive, as stated by the authors, quantitative information on the collagen distribution and organization. Overall, the manuscript is well-written and the references are well-chosen. This contribution could be accepted for publication in Photonics if the authors address the following minor comments and questions that could help the reader to get a better analysis and interpretation of the presented results.  

Minor comments:

Line 152-158:

Only for a better consistency of the manuscript, I would give a more detailed definition of the R² parameter as defined in ref 24 and explain why the value of 0.9 is here specifically chosen. I am also wondering if the results of Fig3 are significantly different for values fixed at 0.5, 0.7, 0.8 or 0.95, for instance. This latter point, in particular, must be commented.

Also, it would be interesting for the reader to visually compare the difference between two representative polarization traces/results extracted from pixels that are treated or, conversely excluded for the subsequent analysis. Are the pixels in question the brightest or darkest in figs 1 and 2 ?

Line 188:

 Why Fig1 is in black & white?

Also, comments are needed here: The optical sectioning clearly results in images with different contrasts according to the position of the z-stage.  One can thus anticipate that the polarization treatment is z-dependent and that depolarization might occur according to the depth of analysis due to a higher scattering of the excitation beam and/or SH signals. Is there an optimal position in z and what is the typical sample thickness which is not specified in the “materials and methods” section?

Can we expect such an evolution in the collagen organisation and structure along z or does the change in the SHG image contrast only result from optical scattering? I am also wondering if the results of Figure 3 are z-dependent. If yes, the position must be specified or at least it should be mentioned if the upper, middle, or lower part of the sample is imaged.  

Line 197, I am not a specialist of the SH contrast that can be obtained from collagen rich sample but can the authors comment on the fact that several ROI in the images of figures 1 and 2 lead to low SHG signal whatever the input polarisation is? Is that a signature of the collagen concentration (or orientation) within the sample? In the same line, can we correlate the results of Figure 3 with the SH intensity (or pixel brightness)? If yes, that information should be included in the revised version.

Line 260, check the section numbering, the conclusion is the 4th section.  

Major comments:

The main message of the manuscript is that the general and trigonal models lead to better fitting results which is obvious, from the mathematical point of view, since the number of parameters increase from the cylindrical model to the general one.

The rotation step of the incoming linear polarization is 10°. Is this parameter can be optimized? One can expect a change in the mean ratio parameter if the angle is reduced for instance.

One can reasonably wonder if the mathematical approach here developed is consistent with the physical and biological parameters we can derive from the collagen structure and organisation in tissue. I mean, regarding the number of independent coefficients within the chi2 tensor, for instance, there is an interesting study (from ensemble measurements and not microscopy images, Measurement of the Second-Order Hyperpolarizability of the Collagen Triple Helix and Determination of Its Physical Origin, JPC B 2009) which could be cited and discussed.

The main question here is the following. Does the use of a low-symmetry model such as the trigonal one can help to derive more quantitative information as stated by the authors and which information (mis-orientation, bending, composition variations, optical scattering or absorption, ..)?

Reviewer 2 Report

The manuscript "Comparison between cylindrical and trigonal symmetry models for the analysis of polarization-dependent second-harmonic generation measurements acquired from collagen-rich biological samples" by M. Mari et al presents a comparison of different collagen models used for the PSHG datasets fitting. Although the topic might be of interest for researchers working with PSHG microscopy, the three considered collagen models are compared only from one parameter perspective: the ratio parameter. Moreover a similar idea of comparing different collagen models was earlier proposed in "Paun, Bogdan, et al. "Strategies for optimizing the determination of second-order nonlinear susceptibility tensor coefficients for collagen in histological samples." IEEE Access 7 (2019): 135210-135219." Here the ratio parameter was called fitting efficiency.

In order to enhance the quality of the paper the authors should consider different types of tissues where different collagen models might be more appropriate. Even more interesting would be to compare different pathologies for the same type of tissue to infer a possible change of collagen structure with pathology.

Moreover, the different model should be compared not just based on the ratio parameter but also considering different other aspects. For example, even if the trigonal symmetry might provide better result, the processing time is 4 orders of magnitude higher compared to the DFT for cylindrical symmetry (the authors should revise their statement in line 120). Other parameters for comparison are welcome.

After extending the considered targeted tissue types/pathologies, the authors might also address the following points:

  1. There are some inconsistencies in the manuscript starting with the title where the authors mention the comparison of two models, while 3 are considered in the manuscript. Moreover, in lines 19-22 in the abstract, on one hand the general model is supposed to provide the best outcome, in the next line, the "findings suggest that trigonal symmetry model describes more precisely the PSHG datasets".
  2. In line 35, I think that the authors are referring to collagen fibrils.
  3. In line 93 the authors should provide full details of the collection objective. Why was not also the BSHG signal considered?
  4. In line 170 it is not clear how the algorithm for trigonal symmetry implemented. Does it also use the FFT algorithm proposed for cylindrical symmetry?
  5. I don't understand the rationale behind Fig. 1 where the authors show depth dependent SHG images while these are not the subject of the paper.
  6. In fig 2 the vertical scale shows a maximum value of 8 which I find very low if it is either counts per second or pixel values. The authors should clarify this aspect
  7. Ref 29 is not suitable to demonstrate the B value for pericardium since in that reference the cornea and tendon are studied
  8. In line 219, why was the cylindrical symmetry tested with both DFT and fitting, since the DFT performance was previously demonstrated?
  9. There is also one result which should be thoroughly explained. For cylindrical symmetry the mean value for B is 1.57, while for trigonal symmetry this value is 1.98. Since such a consistent difference between the two is obtained, how can the two models be compares only by the ratio parameter.

Round 2

Reviewer 1 Report

The revised manuscript is enough consistent.

can be published after minor syntax correction and text editing, especially in lines 174-181 

Reviewer 2 Report

The authors have partially addressed the issues identified in the previous review report. I will discuss their answer point by point. 

  1. ok, although I do not fully agree with the statement in lines 21-23. Even though the trigonal model outperforms the cylindrical one in terms of number of pixels fitted with R2 > 0.9, the computation time is 4 orders of magnitude higher: seconds for FFT-based implementation of cylindrical model and hours for the trigonal model. I can simply imagine an experiment when just by doubling the number of images you get higher number of pixels with R2 > 0.9 with the cylindrical model that with the trigonal model in a shorter time. And this taking into account also the acquisition time (less that one second per image as presented in the manuscript).
  2. ok
  3. ok
  4. ok
  5. I still don't understand why present a 3D stack since it is not used for data analysis. 
  6. ok
  7. If one considers type I collagen in general, than any reference from the current reference list dealing with B calculation from SHG images would have been appropriate. As the authors mention in the answer on point 9 a large spread of values is reported: 1.1-2.6. I was suggesting to present a previous reference with measurement on pericardium. If this is not available, than the authors should mention that their report is the first one with such data on pericardium.
  8. ok
  9. I don not agree with the authors. Even though a wide range of values is reported in literature for type I collagen in different tissues and pathologies, these data were reported using a single collagen model per paper. I also don't agree with the authors when they claim that a change from 1.57 to 1.98 to be just "a slight difference in the calculated anisotropy parameter values for the two models". A 20% change in B is a consistent change. To name just an example, in "Tokarz, D., Cisek, R., Joseph, A., Asa, S. L., Wilson, B. C., & Barzda, V. (2020). Characterization of pathological thyroid tissue using polarization-sensitive second harmonic generation microscopy. Laboratory Investigation100(10), 1280-1287." a change of less than 1% from normal to pathological is considered statistically significant (see table 1 in the reference). The change reported here when using different collagen models should be thoroughly explained.

Also, the main problem which I find here is related with the relevance of the manuscript. I think that the result were obvious from the beginning. For me it is like fitting experimental data with polynomial functions of higher degree, hence higher number of free parameters. It is obvious that there will be a higher number of cases with higher R2 for higher degree polynomial functions. This is also the case here. If tested only on one sample, collagen models with higher number of free parameters will respond better in terms of number of pixels with R2 > 0.9.

My proposal in the initial report was to test these 3 different collagen models on different tissues and/or different pathologies. The conclusion drawn in this manuscript that one model might be better just in term of number of pixels with R2 > 0.9 can hardly be generalized. To be more precise, the results here contradict the results in ref 9 in this manuscript. In ref.9 for breast tissue, the cylindrical symmetry was more suitable for normal tissue, while trigonal symmetry for malignant tissue.

To conclude, the results presented here were obvious, and the conclusion of the manuscript that one collagen model or the other might be better can not be generalized just based on results obtained on one tissue sample.

Reviewer 3 Report

All concerns and suggestions have been addressed. Therefore, I support the publication.

Author Response

Thank you for your positive response.